# Preparation and Performance Study of Radiation-Proof Ultra-High-Performance Concrete

**Min Zhou [1], Tengyu Yang [1], Jinhui Li [2],\*, Bing Qiu [1], Wenjun Qiu [1], Baiyun Li [1], Benan Shu [1], Jinhua Gong [3], Lixian Guo [1] and Yongling Li [1]**

1 Foshan Transportation Science and Technology Co., Ltd., Foshan 528000, China; shuba0411@126.com (M.Z.); yangtengyu@126.com (T.Y.); 13902598673@139.com (B.Q.); 17620132531@163.com (W.Q.); libaiyun1996@163.com (B.L.); shuba@whut.edu.cn (B.S.); jumo731374828@163.com (L.G.); lyling19941209@163.com (Y.L.)
2 College of Materials Science and Engineering, Wuhan Textile University, Wuhan 430200, China
3 School of Materials and Science Engineering, Wuhan University of Technology, Wuhan 430070, China; gjh15779668331@163.com
\* Correspondence: lijinhui-2001@163.com

**Abstract:** With the continuous development of nuclear technology, it is necessary to urgently solve the nuclear safety problem. γ-rays have a strong penetrating power. The γ-ray-shielding performance of ordinary concrete prepared with natural sand is weak and cannot meet the practical application of engineering. The γ-ray shielding performance of concrete can be effectively improved through the use of titaniferous sand with a better γ-ray protection effect. To prepare ultra-high-performance concrete (UHPC) that can provide radiation protection, the influence law of its performance was investigated. The effects of ilmenite sand on the workability, mechanical properties, durability, and radiation-shielding properties of UHPC were investigated via mix testing, compressive strength and flexural strength testing, and a radiation-shielding simulation. The results show that an appropriate amount of ilmenite sand can improve the ultra-high-performance concrete's work performance; however, ilmenite sand has little effect on its compressive strength. Although it is not conducive to the development of flexural and tensile strength, the γ-ray shielding performance of the UHPC increases with an increase in the addition of ilmenite sand. When the titanite sand admixture is 70%, the γ-ray linear absorption coefficient of the prepared UHPC is 0.158 cm$^{-1}$, and the γ-ray shielding performance is significantly improved; meanwhile, its durability performance is excellent.

**Keywords:** ultra-high-performance concrete; titaniferous sand; γ-ray shielding performance

## 1. Introduction

Nuclear technology has been developing very rapidly in recent years [1], and the use of nuclear energy for nuclear power plants [2,3], medical care [4], agriculture [5], and space research [6] is increasing. To provide safe working conditions for nuclear technology, the use of protective measures is necessary [7]. As the most widely used radiation shielding material [8,9], concrete has many advantages: it has a low cost, is easy to mold, and can be widely sourced [10–12]. However, the radiation protection performance of ordinary concrete is poor, and experts have been trying to improve the radiation protection properties of concrete. The energy of a γ-ray is higher than 124 keV, and its short wavelength and strong penetrating ability make it a type of ray with a high shielding-difficulty factor. In the course of their research, researchers found that heavy concrete (volume weight $\geq$ 2600 kg/m$^3$) [13,14] has superior γ-ray-shielding properties when compared to ordinary concrete [15]. The gamma-radiation shielding effect depends mainly on the density, thickness, and atomic number of the absorbing material [16,17]. The percentage of aggregates in concrete is about 50%. Aggregates containing heavy elements can be used to prepare dense concrete with excellent radiation protection properties with

little or no effect on performance. High-density aggregates such as magnetite, ilmenite, hematite, and barite are used as aggregates in concrete [18–23]. The results show that the gamma-ray-shielding performance of concrete mixed with heavy elemental aggregates is significantly improved. Akkurt [24] and Demir [25] believed that with the increase in barite content, the linear attenuation coefficient of a γ-ray in concrete also increased. The results from Ouda [26] showed that compared with concrete containing barite, goethite, and serpentine, the physical and mechanical properties of high-performance and high-density concrete containing magnetite as a fine aggregate were improved, and the shielding efficiency toward γ-rays was improved. Kharita [27] used different types of gamma rays (Cs-137 and Co-60 sources) to test concrete with different aggregates. The results showed that the half-value layer thickness (HVL) of concrete prepared with hematite was 10% lower than that of ordinary concrete. Han [28] found that ilmenite has a high density and a high content of heavy metal elements which are suitable for preparing gamma-ray-shielding concrete.

The design strength of ordinary radiation-proof concrete is low, the density of a heavy aggregate is large, and the difference between the composition of the cementitious materials is easy to segregate [29]. This applies to conventional medical treatment, basic theoretical research, etc., but not to some special projects, such as nuclear power plants. This is because these projects may also face problems such as earthquakes, accidental explosions, weapon strikes, etc. [30]. Therefore, concrete used in this application must have not only good radiation protection performance but also strong bearings and an explosion-proof ability. Therefore, there is a requirement to develop a radiation-proof concrete material with excellent mechanical properties. UHPC is used as a concrete material with ultra-high strength and ultra-high toughness [31]. It is the "material of the future" and can respond to the needs of most buildings. However, due to its high cost, it has not been accepted well in most countries [32]. Meanwhile, relatively little research has been conducted on UHPC in the field of radiation protection, and the application prospects of radiation-proof UHPC are very broad.

Due to a large amount of cementing material and a large amount of hydration heat release, UHPC easily produces cracks and other defects in the process of use, and radiation may radiate along the defects, affecting the radiation protection effect of the UHPC. To improve these problems, the microstructure of UHPC was improved through the use of high titanium heavy slag sand as the reference sand for the "internal maintenance aggregate", and a different proportion of ilmenite sand (apparent density: 4000 kg/m$^3$) was used to replace the high titanium heavy slag sand to enhance the radiation protection effect of the UHPC. The test results can be used as a reference for high-strength radiation protection materials in the medical field and the fields of electric power and nuclear power.

## 2. Experimental Program

### 2.1. Raw Materials

The main raw materials used in this test were Portland cement (P●II52.5 cement), fly ash microbeads (specific surface area: 1300 m$^2$/kg), silica fume (specific surface area: 21,500 m$^2$/kg), swelling agent (7d restricted swelling rate: 0.062% in water), high-efficiency-type polycarboxylic acid water-reducing agent (solid content 50%; water reduction rate of up to 35%), copper-plated short straight steel fiber (length: 12~14 mm; diameter: 0.23 mm), water, high titanium heavy slag sand (density: 3100 kg/m$^3$, The gradation is shown in Figure 1), ilmenite sand (the particle size was 3 mm, the TiO$_2$ content was 52%, the Mohs hardness was 6, and the density was 4000 kg/m$^3$).

The specific compositions of the cement, silica fume, and fly ash microbeads used in this study are shown in Table 1.

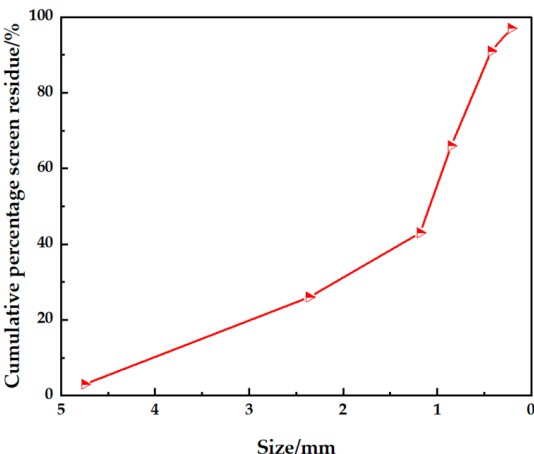

**Figure 1.** Gradation of high titanium heavy slag sand.

**Table 1.** Chemical composition of cementitious materials/wt%.

| Sample | $SiO_2$ | $Al_2O_3$ | $Fe_2O_3$ | CaO | MgO | $SO_3$ | $K_2O$ | $Na_2O$ | $TiO_2$ |
|---|---|---|---|---|---|---|---|---|---|
| Cement | 19.83 | 4.38 | 3.27 | 64.08 | 1.25 | 2.21 | 0.84 | 0.16 | 0.25 |
| Silica fume | 93.05 | 0.48 | 0.13 | 0.79 | 0.33 | 0.75 | 0.14 | 0.07 | 0 |
| Fly ash microbeads | 62.6 | 19.71 | 3.72 | 6.28 | 0.97 | 0.13 | 1.36 | 1.22 | 0.67 |

*2.2. Test Methods*

The working properties were tested according to The Standard for Test Methods for Properties of Ordinary Concrete Mixes (GB/T 50080-2016). The mechanical properties were tested using a universal testing machine in accordance with The Standard for Test Methods for Mechanical Properties of Ordinary Concrete (GB/T 50081-2002) (compression test block: 100 mm × 100 mm × 100 mm and flexural test block: 100 mm × 100 mm × 400 mm, each in a group of three). The radiation protection performance was simulated and analyzed using the Monte Carlo method. The durability performance was tested using The Standard for Long-term Performance and Durability Test Methods for Ordinary Concrete (GB/T 50082-2009) (mainly testing the performance of resistance to chloride ion erosion and resistance to carbonization).

The linear attenuation coefficient of γ-rays was calculated according to Lambert's Law [33]. The half-value layer (HVL) and the ten-value layer (TVL) correspond to the thicknesses required for the UHPC to absorb one-half and one-tenth of the incident ray intensity relative to the original value, respectively. The specific calculation method is as follows.

$$I = I_0 e^{-\mu t} \tag{1}$$

$$HLV = \frac{ln2}{\mu} \tag{2}$$

$$TVL = \frac{ln10}{\mu} \tag{3}$$

where $I$ is the initial intensity of the ray; $I_0$ is the residual intensity of the ray through the sample; $\mu$ is the linear attenuation coefficient/neutron absorption cross-section ($cm^{-1}$); t is the thickness of the concrete (mm); $HLV$ is the thickness of half-value layer (cm); $TVL$ is the thickness of the ten-value layer (cm).

*2.3. Mixing Ratio Design*

Ilmenite sand is a high-density aggregate and contains 35% iron, which can effectively absorb gamma rays. Thus, ilmenite sand is a good material for radiation protection. In this

experiment, different doses of ilmenite sand were used to replace high titanium heavy slag sand, and the ilmenite sand was replaced by 0, 10%, 30%, 50%, and 70% by volume (A0, A1, A2, A3, and A4) to study the effect of different doses of ilmenite sand on the working performance and mechanical properties of UHPC. Specific fits are shown in Table 2.

**Table 2.** Different ilmenite sand blending test ratios (kg/m$^3$).

| Group | Cement | Silica Fume | Fly Ash Microbeads | Water | Water Reducing Agent | Expansion Agent | High Titanium Heavy Slag Sand | Ilmenite Sand | Steel Fiber |
|---|---|---|---|---|---|---|---|---|---|
| A0 | 767 | 180 | 180 | 204 | 30 | 72 | 1100 | - | 200 |
| A1 | 767 | 180 | 180 | 204 | 30 | 72 | 990 | 141.9 | 200 |
| A2 | 767 | 180 | 180 | 204 | 30 | 72 | 770 | 425.7 | 200 |
| A3 | 767 | 180 | 180 | 204 | 30 | 72 | 550 | 709.5 | 200 |
| A4 | 767 | 180 | 180 | 204 | 30 | 72 | 330 | 993.3 | 200 |

## 3. Results and Discussion

### 3.1. Mix Performance

The radiation protection performance of the UHPC mix mainly the includes slump, expansion, and volume weight; the volume weight has a greater impact on radiation shielding, while the slump and expansion are not only important indicators of work performance but can also affect the radiation protection effect to a certain extent. The test results of the UHPC mix's performance are shown in Table 3 and Figure 2.

**Table 3.** Effect of ilmenite sand admixture on the performance of UHPC mixes.

| Group | Slump/mm | Expansion Degree/mm |
|---|---|---|
| A0 | 265 | 630 |
| A1 | 270 | 640 |
| A2 | 265 | 645 |
| A3 | 280 | 655 |
| A4 | 260 | 630 |

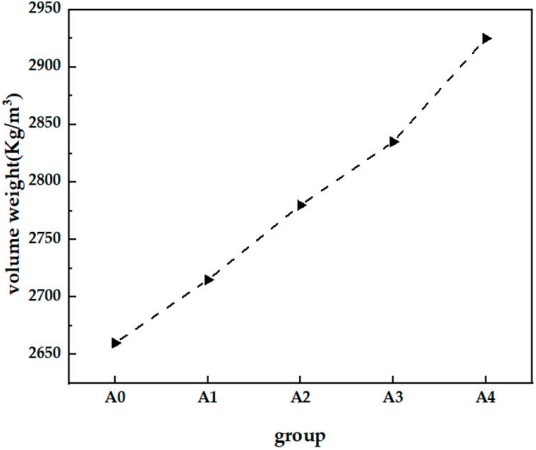

**Figure 2.** Effect of ilmenite sand admixture on the performance of UHPC mixes.

From the test results in Table 3 and Figure 2, it can be noted that with the increase in the ilmenite sand admixture, the slump and expansion of the UHPC show a trend of increasing first and then decreasing. When the volume of the admixture of ilmenite sand is 50%, the slump/expansion reaches 280 mm/655 mm. This is because ilmenite sand is a spherical, fine aggregate with a smooth surface, regular shape, and uniform particle size, factors which have a good effect on its dispersion in the system when added in appropriate amounts,

reducing the viscosity between slurries [34] and facilitating the improvement of the UHPC's working performance. However, when it is mixed in excess, its excessive surface area, due to its small particle size, adversely affects the slump/extensibility. Therefore, the working performance of the A4 group is reduced compared to the A3 group. Meanwhile, it can be seen that the volume weight of the UHPC gradually increased with the increase in the ilmenite sand replacement, and when the amount of ilmenite sand replacing high titanium heavy slag sand reaches 70%, the volume weight of the UHPC reaches 2925 kg/m$^3$. This is because on one hand, ilmenite sand is very dense (about 4000 kg/m$^3$) and has a higher density than high titanium heavy slag sand (3100 kg/m$^3$); on the other hand, ilmenite sand has a finer grain size, which can reduce the large pores and improve the density to a certain extent. Therefore, with the increase in the ilmenite sand admixture, the improvement in the volume weight is more obvious.

### 3.2. Mechanical Properties

The results of the test of the effect of ilmenite sand on the compressive and flexural tensile strengths of UHPC are shown in Figure 3.

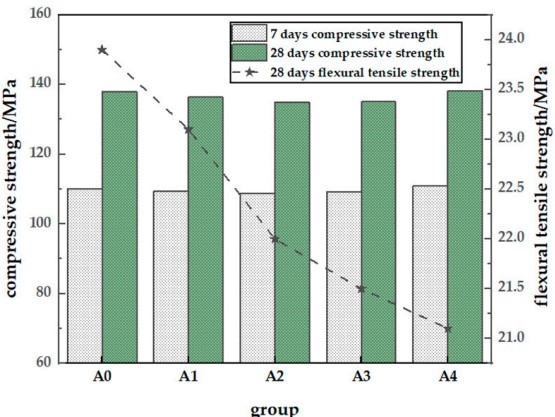

**Figure 3.** Effect of ilmenite sand admixture on the mechanical properties of UHPC.

It can be seen from Figure 3 that with the increase in the titaniferous sand admixture, the 7-day and 28-day compressive strengths of each group of UHPC do not demonstrate significant changes, and the 28-day compressive strength is in the range of 135~140 MPa. This is because to a large extent, the compressive strength of concrete is related to the characteristics of the aggregate, and ilmenite sand and high titanium heavy slag sand are high-strength aggregates; the impact of their effect on the compressive strength of the concrete is not very different. However, the 28-day flexural tensile strength gradually decreased with the increase in the titaniferous sand admixture, and when the admixture amount reached 70% (group A4), the 28-day flexural tensile strength of the UHPC was only 21.1 MPa, and the flexural strength of group A0 decreased by 10.33%. This is because the bending and tensile strength is affected by a different mechanism than the compressive strength. It mainly relates to the matrix of the cementitious material. The internal maintenance effect of the water-saturated pre-wetted high titanium heavy slag sand and the "pin structure" formed with the cementitious material effectively improve the interfacial transition zone (ITZ) of the UHPC; it reduces the thickness of the ITZ, and it improves the strength of the transition zone so that it can resist the bending action without being destroyed [35]. The macroscopic performance shows that the flexural tensile strength of the UHPC gradually decreased with the decrease in the high-titanium heavy mineral sand doping. Considering these properties, the recommended admixture of ilmenite sand is 50%.

### 3.3. Radiation Protection Performance

The provision of shielding protection from radiation fields is an issue that needs to be addressed in future work. Therefore, it is of great importance to carry out ray-shielding experiments. Considering that radiation-shielding experiments require special sites, specialized personnel, and very expensive equipment, they can cost a significant amount of money and time. The radioactive substances produced at the same time can also bring a series of problems. With the rapid development of computer technology, a virtual simulation can be a good solution to this problem. In this experiment, the gamma-ray-shielding abilities of UHPCs prepared with different doping amounts of ilmenite sand are simulated and studied.

The simulation process, combined with the chemical compositions of raw materials with different UHPC ratios, calculated the mass percentage of each element in the UHPC, as is shown in Table 4.

**Table 4.** Mass percentage of each element of UHPC (%).

| Group | Si | Al | Fe | Ca | Mg | S | K | Na | Ti | H | C | O |
|-------|-------|------|-------|-------|------|------|------|------|-------|------|------|-------|
| A0 | 12.00 | 4.92 | 11.65 | 22.47 | 0.30 | 0.31 | 0.31 | 0.67 | 5.81 | 0.88 | 0.39 | 40.28 |
| A1 | 11.45 | 4.51 | 13.15 | 21.37 | 0.29 | 0.30 | 0.31 | 0.60 | 6.88 | 0.87 | 0.38 | 39.88 |
| A2 | 10.41 | 3.73 | 16.00 | 19.28 | 0.28 | 0.29 | 0.30 | 0.48 | 8.90 | 0.84 | 0.37 | 39.11 |
| A3 | 9.43 | 2.99 | 18.68 | 17.31 | 0.27 | 0.28 | 0.29 | 0.36 | 10.81 | 0.82 | 0.36 | 38.39 |
| A4 | 8.51 | 2.29 | 21.21 | 15.46 | 0.27 | 0.28 | 0.28 | 0.25 | 12.61 | 0.79 | 0.35 | 37.71 |

The concrete test block size was set to a diameter of 70 mm and a thickness of 0~10 cm, and the radiation attenuation data at 3 cm, 5 cm, 10 cm, and 15 cm thicknesses were measured to determine the shielding coefficient of the UHPC. The distance between the point source and the front surface of the concrete was set to 50 cm.

As can be seen from Figure 4, the γ-ray absorption coefficient of the UHPC increased gradually from 0.146 to 0.158 with the increase in the UHPC's titanium iron ore content. This situation shows that the incorporation of ilmenite sand can effectively improve the γ-ray-shielding performance of UHPC, and the γ-ray shielding effect of UHPC increased with the increase in ilmenite sand content in the UHPC.

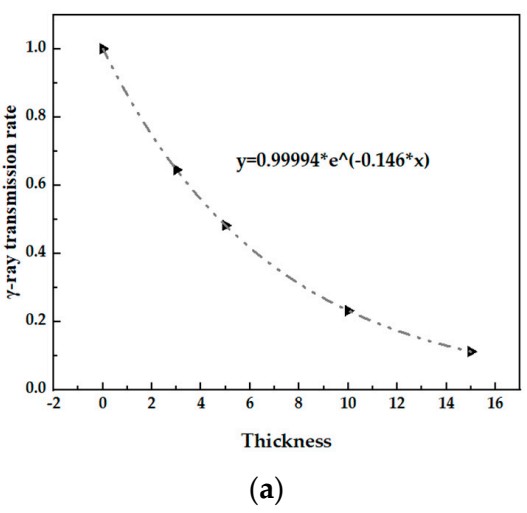

**(a)**

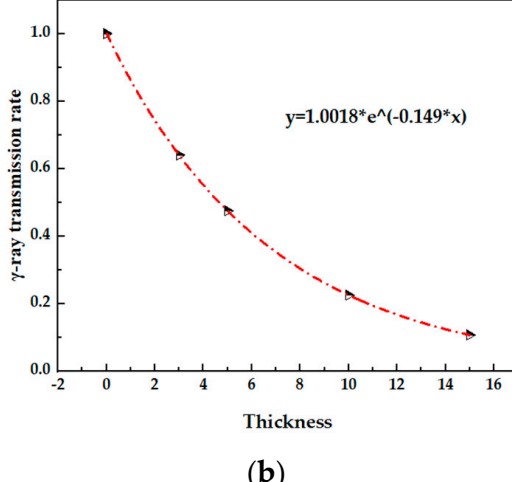

**(b)**

**Figure 4.** *Cont.*

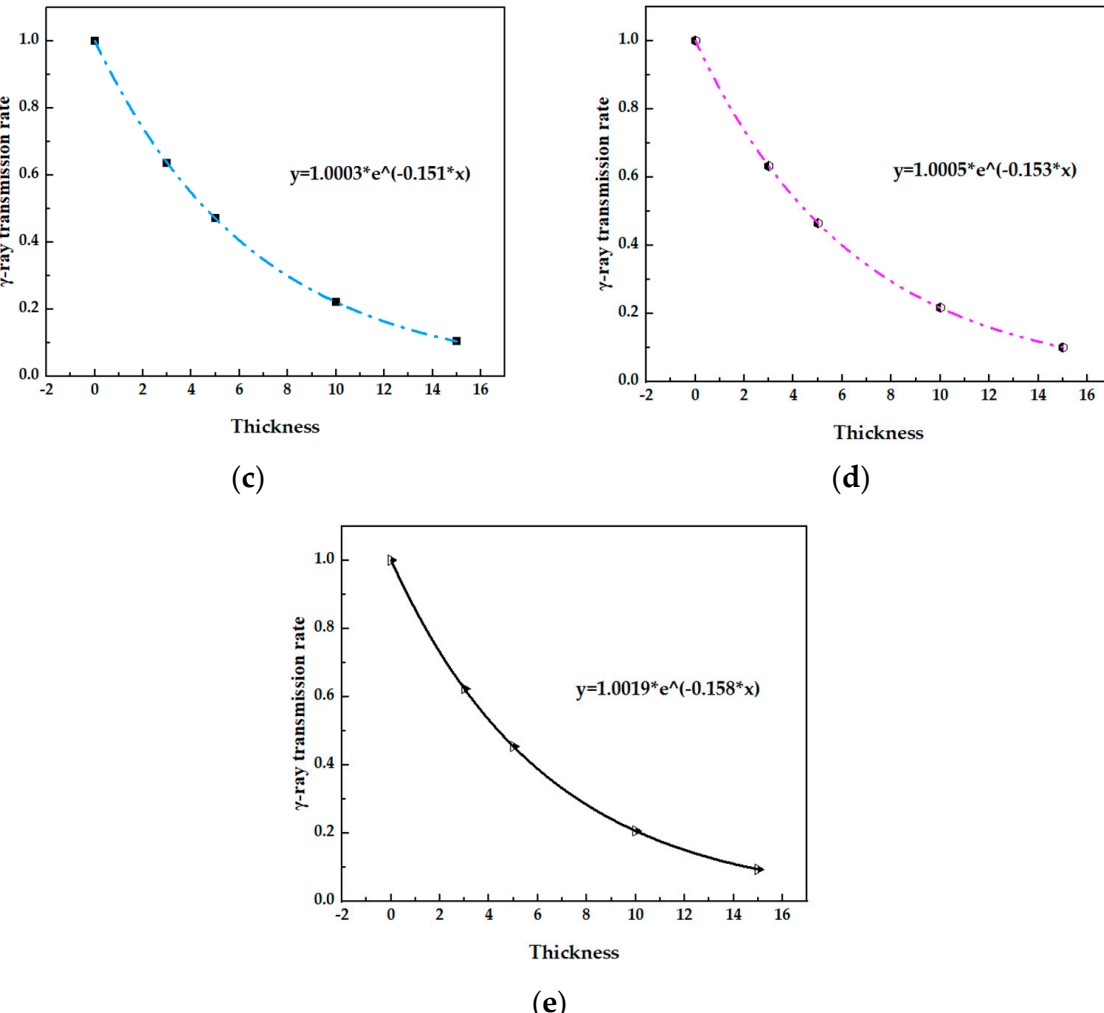

**Figure 4.** UHPC γ-ray transmission curves of different thickness groups. (**a**) A0; (**b**) A1; (**c**) A2; (**d**) A3; (**e**) A4.

It can be seen from Figure 4 and Table 5 that under the same conditions, the γ-ray-shielding performance increased with the increase in ilmenite sand content. Compared with group A0, the half-value layer of group A4 was reduced by 7.6%, and the ten-value layer was reduced by 5.7%. This is because the apparent density of ilmenite sand is 4000 kg/m$^3$, which belongs to the category of heavy aggregates [15], and ilmenite sand contains a large number of heavy elements such as titanium and iron [36]. Under the combined actions of these two factors, the γ-ray-shielding performance of UHPC increased with the increase in ilmenite sand content.

**Table 5.** Simulation results of UHPC γ-ray shielding attenuation for each group.

| Group | Linear Absorption Coefficient/cm$^{-1}$ | HVL/cm | TVL/cm |
|:---:|:---:|:---:|:---:|
| A0 | 0.146 | 4.75 | 15.77 |
| A1 | 0.149 | 4.65 | 15.45 |
| A2 | 0.151 | 4.59 | 15.25 |
| A3 | 0.153 | 4.53 | 15.05 |
| A4 | 0.158 | 4.39 | 14.57 |

### 3.4. Durable Performance

UHPC has the advantages of a dense structure and small internal pores, so its durability is relatively large. The radiation-proof UHPC or its long-term service (exposure to

radiation) will certainly have a certain impact on its durability, so it should have a higher requirement for durability. In this section, we investigate the durability of radiation-proof UHPC, such as its resistance to chloride ion penetration and carbonization.

### 3.4.1. Anti-Chlorine-Ion Penetration Performance

To a certain degree, concrete's resistance to chloride ion erosion can directly reflect its durability. In general, concrete with good chloride ion attack resistance has excellent durability [37]. This study used the electric flux method to test the chloride ion penetration resistance of the UHPC. However, the incorporation of steel fibers will have a certain impact on the test data of the electric flux method, and this section selected the A0~A4 group with the removal of steel fibers to test the UHPC's performance with respect to its resistance to chloride ion attack, and the test results are shown in Table 6.

**Table 6.** Anti-chlorine-ion penetration test results.

| Group | Electric Flux (C) | Electric Flux Rating Level |
|---|---|---|
| A0 | 44 | Qs-V |
| A1 | 51 | Qs-V |
| A2 | 59 | Qs-V |
| A3 | 64 | Qs-V |
| A4 | 77 | Qs-V |

As can be seen from Table 6, the radiation-proof UHPC demonstrated good resistance to the penetration of chloride ions, and the electric flux of each group was $\leq$500 C, which can reach the Qs-V level and is much lower than 1300 C of ordinary C50 concrete. Meanwhile, the electric flux of the UHPC gradually increased with the increase in the boron carbide admixture. To analyze the reason, UHPC has a low water/cement ratio, and it is mixed with cementitious materials such as fly ash and silica fume, with no coarse aggregate and a high level of compactness, which can hinder the diffusion of chloride ions in its interior to a certain extent. In addition, high titanium heavy slag sand pre-wetted with full water in advance can promote the hydration of cementitious materials, improve the internal pore structure of neutron-shielded UHPC, and subsequently enhance its denseness. In contrast, ilmenite sand has no internal conservation effect and relatively more chloride ion transport channels. Therefore, with an increase in the amount of titaniferous sand, the UHPC demonstrates relatively poor chloride ion penetration resistance, but its electric flux rating level is still at the Qs-V level, with excellent resistance to chloride ion erosion.

### 3.4.2. Carbonation Resistance

Carbonation is a process of chemical corrosion and is mainly the neutralization reaction of calcium carbonate formed through the reaction of hydration products and carbon dioxide. During the carbonization process, the PH value inside the concrete is reduced to <9.0 (destroying the alkaline environment), which leads to the corrosion of the steel bars inside the concrete and destroys the dense structure of the concrete. As a radiation-shielding structure, radiation-proof UHPC needs to bear the load and maintain the stability of the structure in addition to shielding from gamma rays. In this study, specimens were exposed to carbonization in a carbonization chamber with a humidity of $75 \pm 5\%$, a temperature of $20 \pm 2\ ^{\circ}C$, and a $CO_2$ gas concentration of $20 \pm 3$ for up to 3, 7, 14, and 28 days, and the carbonization depth was measured to study the volumetric stability of the radiation-proof UHPC. The 28 d carbonation depth of the UHPC is shown in Table 7.

It can be seen from the data in Table 7 that the carbonization depth of the UHPC in each group was very low, and the carbonization depth of the A0~A4 groups was 0 mm with an increase in age. This indicates that the UHPC prepared via the "internal curing" of high titanium heavy slag sand and ilmenite sand has fewer interconnected pores and a high level of compactness, which can effectively inhibit the diffusion of carbon dioxide

inside UHPC [35]. It can be observed that the prepared UHPC demonstrated excellent resistance to carbonization.

**Table 7.** Carbonation resistance test results.

| Group | The Carbonation Depth of UHPC at Different Ages/mm | | | |
|---|---|---|---|---|
| | 3 Days | 7 Days | 14 Days | 28 Days |
| A0 | 0 | 0 | 0 | 0 |
| A1 | 0 | 0 | 0 | 0 |
| A2 | 0 | 0 | 0 | 0 |
| A3 | 0 | 0 | 0 | 0 |
| A4 | 0 | 0 | 0 | 0 |

*3.5. Microstructure*

In this study, we prepared UHPC mortar and used SEM and nanoindentation testing to reveal the mechanism of the internal maintenance effect of high titanium heavy slag sand on the microstructure and properties of UHPC.

It can be seen from Figure 5a that the high titanium heavy slag sand is an irregularly shaped porous aggregate with a small ITZ thickness. For this reason, high titanium heavy slag sand that is pre-wetted with water has many micro-fine interconnecting pores with an internal conservation effect [35,38]. In the hydration process, the internal moisture is continuously released outward under the action of a capillary force, which improves the hydration degree, improves the structure of the ITZ, reduces the thickness of the ITZ, and enhances the bonding degree between the aggregate and the cementitious slurry. The ITZ between the ilmenite sand and cementitious material is more obvious in Figure 5b. This is because ilmenite sand is a type of regularly shaped and dense aggregate, and the degree of bonding between it and the cementitious material is poor because there is no internal maintenance effect. However, ilmenite sand itself is a high-strength aggregate, so the compressive strength of the UHPC does not change much as the amount of ilmenite sand increases; however, the bending and tensile strength gradually decrease.

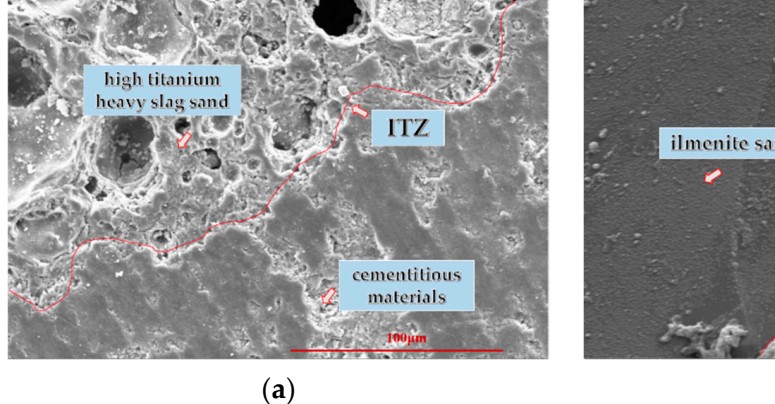
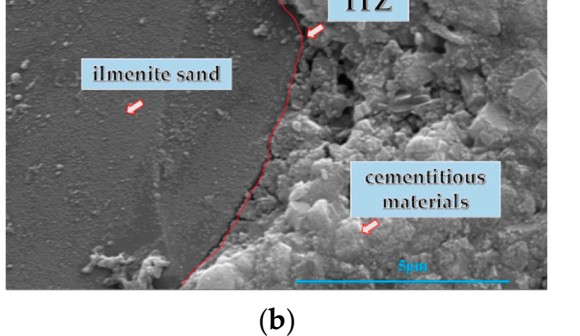

(**a**)　　　　　　　　　　　　　(**b**)

**Figure 5.** Effect of different aggregates on the microstructure of UHPC. (**a**) High titanium heavy slag sand; (**b**) ilmenite sand.

Table 8 Shows the results of nanoindentation tests of high titanium heavy slag sand and quartz sand. It can be seen from the table that the average elastic modulus of the interfacial transition zone of the high titanium heavy slag sand is up to 28.33 GPa, while that of quartz sand is only 24.58 GPa. In addition, it can be found that although it is porous, the high titanium heavy slag sand has excellent properties of its own, with an elastic modulus and hardness 1.4 and 2 times those of the quartz sand, respectively. This is because the nano-mechanical properties of the interfacial transition zone and the matrix

are more significantly improved than those of quartz sand under the internal maintenance effect of high titanium heavy slag sand.

**Table 8.** UHPC nanoindentation test results.

| Aggregate Distance/mm | High Titanium Heavy Slag Sand | | Quartz Sand | |
| --- | --- | --- | --- | --- |
| | Elastic Modulus/GPa | Hardness/GPa | Elastic Modulus/GPa | Hardness/GPa |
| −10 | 155.56 | 11.22 | 119 | 6.49 |
| −5 | 182.16 | 15.17 | 132.4 | 7.56 |
| 0 | 136.60 | 14.29 | 96.1 | 5.46 |
| 5 | 33.62 | 1.21 | 29.3 | 0.71 |
| 10 | 22.05 | 2.14 | 34.1 | 1.38 |
| 15 | 29.30 | 1.52 | 29.9 | 0.85 |
| 20 | 46.45 | 2.12 | 23.8 | 1.27 |
| 25 | 28.5 | 2.28 | 27.0 | 1.03 |
| 30 | 48.8 | 1.48 | 26.6 | 0.66 |

## 4. Conclusions

(1) As the amount of ilmenite sand replacing high titanium heavy slag sand increases, the compressive strength of the UHPC does not change much, and the flexural strength gradually decreases. The working properties, apparent density, and the strength against γ-rays gradually increase. When the amount of replacement of ilmenite sand is 50%, the 28-day compressive strength of the UHPC is 136.1 MPa, the 28-day flexural strength is 21.6 MPa, and the linear absorption coefficient of γ-rays is 153 cm$^{-1}$.

(2) The internal maintenance effect of the high titanium heavy slag sand can improve the microstructure of the UHPC, increase the degree of hydration, and improve the compactness. The macroscopic performance improves the mechanical properties and durability performance of the UHPC. The addition of ilmenite sand can improve the γ-ray-shielding performance of UHPC. The joint action of the two aggregates can improve their mechanical properties and radiation-shielding performance at the same time.

(3) The properties of the radiation-proof concrete prepared via this preparation technique are a 28 d compressive strength $\geq$ 135 MPa, a flexural strength $\geq$ 20 MPa, a γ-ray linear absorption coefficient $\geq$ 0.150 cm$^{-1}$, and excellent durability performance. The excellent properties of the radiation-protected UHPC prepared in this study can provide theoretical references for future research and engineering applications of radiation-protected concrete.

**Author Contributions:** Methodology, data curation, and writing—original draft, M.Z.; methodology, funding supervision, and writing—review and editing, J.L.; methodology and data curation, T.Y.; methodology and data curation, B.Q.; methodology and data curation, W.Q.; methodology and data curation, B.L.; methodology and data curation, B.S.; methodology, data curation, and software, J.G.; project administration and resources, L.G.; and data curation, Y.L. All authors have read and agreed to the published version of the manuscript.

**Funding:** This work was supported by the Science and Technology Project of Foshan Transportation Science and Technology Co., Ltd. (FJK (2020) B-062).

**Institutional Review Board Statement:** Not applicable.

**Informed Consent Statement:** Not applicable.

**Data Availability Statement:** The data presented in this study are available upon request from the corresponding author.

**Conflicts of Interest:** The authors declare no conflict of interest.

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
