# Peer review of "Preparation and Performance Study of Radiation-Proof Ultra-High-Performance Concrete"

_coatings, doi:10.3390/coatings13050906_

Round 1

Reviewer 1 Report

Dear Authors

All comments in the attached pdf should be completed . 

Author Response

Dear editor,

Thank you for your letter and for the reviewers’ comments concerning our manuscript entitled “Effect of ilmenite sand on the mechanical performance and gamma-ray shielding performance of radiation-proof UHPC” (ID: coatings-2375986). These comments are all valuable and very helpful for revising and improving our paper. We have studied comments carefully and have made correction which we hope to meet with approval. The main corrections in the paper and the responds to the reviewer’s comments are listed one by one in the following for your consideration.

With best regards,

Jinhui LI

Comment 1:The gradation curve of two type of sand should be add.

Answer: Your comments are valuable and we have included the grading of the aggregates in our paper.

Comment 2:Any explanation of behavior is documented with the results of previous studies.

Answer: This issue was not well thought out and we have cited previous studies and revised this section.

Comment 3:The section of conclusion so weak should be rewrite to be more clear.

Answer: We have revised the conclusions according to your comments and reflected them in the paper.

Reviewer 2 Report

1- The article needs grammatical and syntax improvements. The linguist level of this article does not meet the requirements.

2- Title

The title is probably the most crucial section of the whole paper. The title immediately clues the reader and the reviewers into what your point is and why it’s important. The title is not informative, specific, and understandable. The authors should modify the title.

3- Abstract

The abstract requires significant revision to improve the quality of the manuscript.

4- Introduction

4-1- In the introduction section, the literature review part is generally not well organized. The introduction needs to be revised.

4-2- The theoretical, analytical, and standard approaches should be discussed.

5- The following paragraphs (sentences) are unclear:

5-1- This is because, ilmenite sand belongs to the spherical fine aggregate with a smooth surface, regular shape, and uniform particle size,  when it is mixed in the appropriate amount, it has a good dispersion effect in the system, which is conducive to the improvement of UHPC working performance; but when it is mixed in excess, because of its small particle size, excessive surface area, and its poor grading, these have adverse effects on the slump/extension, so the working performance of A4 group is reduced compared with A3 group. Meanwhile, it can be shown that the volume weight of UHPC gradually increases with the increase of ilmenite sand replacment, and when the amount of ilmenite sand replacing high titanium heavy slag sand reaches 70%, the volume weight of UHPC reaches 2925 kg/m3. This is because, on the one hand, ilmenite sand is very dense (about 4000 kg/m3), which is higher than high titanium heavy slag sand (3100 kg/m3); on the other hand, ilmenite sand has a finer grain size, which can reduce the large pores and improve the dense density to a certain extent. Therefore, with the increase of ilmenite sand admixture, the volume weight improvement is more obvious.

5-2- It can be seen from Figure 5. a) that the high titanium heavy slag sand is an irregularly shaped porous aggregate with a small thickness of the ITZ. This is attributed to the reason  that the high titanium heavy slag sand which is pre-wetted with water in advance has  many micro-fine interconnecting pores with an internal conservation effect. In the hydration process, the internal moisture is continuously released outward under the action of  capillary force, which improves the hydration degree, improves the structure of the ITZ,  reduces the thickness of the ITZ, and enhances the bonding degree between the aggregate  …. .

5-3- This indicates that the UHPC prepared by “internal curing” high-titanium heavy slag sand and ilmenite sand has less interconnected pores and high compactness, which  can effectively inhibit the diffusion of carbon dioxide inside UHPC. It can be observed  that the prepared UHPC has excellent resistance to carbonization.

6- Research significance

The purpose of the study is not mentioned in the abstract and introduction. What is the purpose of the study and the contribution of the results to the literature? Also, the significance of the study must be described in a separate section after the introduction.

7- The authors only hardly discuss results. Please discuss your results more deeply.

8- Conclusion

Limited new knowledge can be found in the conclusion part. The conclusion section needs to be re-written.

9- Figures

The descriptions of axis and legends in different figures are not of the same size fonts. Furthermore, authors can enhance the quality of the figures.

10- The format of some references should be checked.

Author Response

Dear editor,

Thank you for your letter and for the reviewers’ comments concerning our manuscript entitled “Effect of ilmenite sand on the mechanical performance and gamma-ray shielding performance of radiation-proof UHPC” (ID: coatings-2375986). These comments are all valuable and very helpful for revising and improving our paper. We have studied comments carefully and have made correction which we hope to meet with approval. The main corrections in the paper and the responds to the reviewer’s comments are listed one by one in the following for your consideration.

With best regards,

Jinhui LI

Reply to reviewer’s comments

Comment 1:The article needs grammatical and syntax improvements. The linguist level of this article does not meet the requirements.

Answer: The grammar of the paper has been revised by your comments.

Comment 2:The title is probably the most crucial section of the whole paper. The title immediately clues the reader and the reviewers into what your point is and why it’s important. The title is not informative, specific, and understandable. The authors should modify the title.

Answer: The title of the article has been changed to "Preparation and performance study of radiation-proof ultra-high performance concrete".

Comment 3:The abstract requires significant revision to improve the quality of the manuscript.

Answer: Your comments were very valuable and we have revised the abstract.

Comment 4:Introduction

4-1- In the introduction section, the literature review part is generally not well organized. The introduction needs to be revised.

4-2- The theoretical, analytical, and standard approaches should be discussed.

Answer: You have made good comments and I have reorganized the language of the review section as requested and discussed the theoretical, analytical, and standard.

Comment 5:The following paragraphs (sentences) are unclear:

Answer: The section has been revised and previous studies have been cited.

Comment 6:The purpose of the study is not mentioned in the abstract and introduction. What is the purpose of the study and the contribution of the results to the literature? Also, the significance of the study must be described in a separate section after the introduction.

Answer: Your points are well taken, and that part is reflected in the article.

Comment 7:The authors only hardly discuss results. Please discuss your results more deeply.

Answer: Your point is well taken, and the section has been revised.

Comment 8:Limited new knowledge can be found in the conclusion part. The conclusion section needs to be re-written.

Answer: We have revised the conclusions according to your comments and reflected them in the paper.

Comment 9:The descriptions of axis and legends in different figures are not of the same size fonts. Furthermore, authors can enhance the quality of the figures.

Answer: The figures has been revised according to your comments.

Comment 10:The format of some references should be checked.

Answer: The format of references has been checked and revised.

Reviewer 3 Report

The manuscript needs language, grammar, and syntactic editing. The English language usage should be checked by a fluent English speaker. It is understood that for the non-English speaker it is ok, but it still needs to be properly checked and proofread it. However, the manuscript, in its present form, contains several weaknesses.

(1) In the review of literature, the latest development should be highlighted rather than piling up the paper.

Recommended references:

:- "Experimental and Numerical Investigations on Performance of Reinforced Concrete Slabs under Explosive-induced Air-blast Loading: A state-of-the-art review". Structures, Elsevier, 31, pp. 428-461, DOI: 10.1016/j.istruc.2021.01.102.

:- "Ultra High Performance Concrete (UHPC) and C-FRP Tension Re-bars: A Unique Combinations of Materials for Slabs subjected to Low-velocity Drop Impact Loading". Frontiers in Materials, Frontiers, DOI: 10.3389/fmats.2022.1061297.

(2) What is the main contribution of this paper? What motives for that? What gap in this field has been covered? Which part of the mentioned instruction has been progressed? According to what criteria?

(3) Some assumptions are stated in various sections. Justifications should be provided on these assumptions. Evaluation on how they will affect the results should be made.

The manuscript needs language, grammar, and syntactic editing. The English language usage should be checked by a fluent English speaker. It is understood that for the non-English speaker it is ok, but it still needs to be properly checked and proofread it. However, the manuscript, in its present form, contains several weaknesses.

Author Response

Dear editor,

Thank you for your letter and for the reviewers’ comments concerning our manuscript entitled “Effect of ilmenite sand on the mechanical performance and gamma-ray shielding performance of radiation-proof UHPC” (ID: coatings-2375986). These comments are all valuable and very helpful for revising and improving our paper. We have studied comments carefully and have made correction which we hope to meet with approval. The main corrections in the paper and the responds to the reviewer’s comments are listed one by one in the following for your consideration.

With best regards,

Jinhui LI

Reply to reviewer’s comments

Comment 1:The manuscript needs language, grammar, and syntactic editing. The English language usage should be checked by a fluent English speaker. It is understood that for the non-English speaker it is ok, but it still needs to be properly checked and proofread it. However, the manuscript, in its present form, contains several weaknesses.

Answer: The grammar of the paper has been revised by your comments.

Comment 2:In the review of literature, the latest development should be highlighted rather than piling up the paper.

Recommended references:

:- "Experimental and Numerical Investigations on Performance of Reinforced Concrete Slabs under Explosive-induced Air-blast Loading: A state-of-the-art review". Structures, Elsevier, 31, pp. 428-461, DOI: 10.1016/j.istruc.2021.01.102.

:- "Ultra High Performance Concrete (UHPC) and C-FRP Tension Re-bars: A Unique Combinations of Materials for Slabs subjected to Low-velocity Drop Impact Loading". Frontiers in Materials, Frontiers, DOI: 10.3389/fmats.2022.1061297.

Answer: According to your opinion, we have revised the summary of the article and referred to the paper you recommended.

Comment 3:What is the main contribution of this paper? What motives for that? What gap in this field has been covered? Which part of the mentioned instruction has been progressed? According to what criteria?

Answer: Your comments are very valuable, and we have supplemented the article according to your comments.

Comment 4:Some assumptions are stated in various sections. Justifications should be provided on these assumptions. Evaluation on how they will affect the results should be made.

Answer: We have revised it according to your opinion, and the specific content can be reflected in the paper.

Reviewer 4 Report

Manuscript ID: coatings-2375986

The subject of the paper corresponds to its content.

The literature review is correct. Source materials are cited mostly from recent years, with the exception of single items.

The research methodology is correct. The standards used in the experiment were indicated. However, there is no reference to the method of carbonation determination.

The experimental setup is correct. In addition to the basic mortar test, which is the compressive strength, flexural tensile strength was also marked. The influence of the used ingredients (mainly the type of aggregate) on these parameters was assessed. On this basis, a significant decrease in flexural strength was observed as the amount of heavy sand increased. The reason for this fact was duly substantiated.

In order to demonstrate the shielding properties of the tested mortars, a study was carried out showing significant protection against gamma-ray. The importance of increasing the amount of heavy sand in the composition of the mortar, resulting in the improvement of shielding properties, was also captured.

Within the scope of the experiment, the behavior of the tested mortars in the chloride environment was also assessed. The presence of heavy sand has not been found to have a beneficial effect on chloride ion penetration, but ultimately it is an acceptable level.

The topic of carbonation was also recognized, but the exact method of the experiment was not described and it is suggested that it be supplemented.

SEM analyzes are a good supplement to the research. Thanks to SEM observations, it was noticed that in the case of heavy sand: slag and ilmenite, the ITZ zone is different, which is influenced by the porous structure of slag sand and a more developed specific surface area of grains. This confirms the generally known theory in this regard.

It is worth noting that the experiment was carried out on cement mortar, not on concrete. It seems that the next stage of research should be carried out on concrete.

To sum up, I propose to supplement the record regarding the research method in the field of carbonation and after this supplementation I recommend the paper for publication.

Author Response

Dear editor,

Thank you for your letter and for the reviewers’ comments concerning our manuscript entitled “Effect of ilmenite sand on the mechanical performance and gamma-ray shielding performance of radiation-proof UHPC” (ID: coatings-2375986). These comments are all valuable and very helpful for revising and improving our paper. We have studied comments carefully and have made correction which we hope to meet with approval. The main corrections in the paper and the responds to the reviewer’s comments are listed one by one in the following for your consideration.

With best regards,

Jinhui LI

Reply to reviewer’s comments

Comment 1:To sum up, I propose to supplement the record regarding the research method in the field of carbonation and after this supplementation I recommend the paper for publication.

Answer: Your opinion is very valuable. We have supplemented the test method of carbonation resistance according to your opinion, and the specific content has been reflected in the paper.

Round 2

Reviewer 3 Report

The authors' have improved the quality of the manuscript. This reviewer recommends publication of this revised manuscript.